

# How well can inverse analyses of high-resolution satellite data resolve heterogeneous methane fluxes? Observation System Simulation Experiments with the GEOS-Chem adjoint model (v35)

Xueying Yu[1], Dylan B. Millet[1], Daven K. Henze[2]

[1]Department of Soil, Water, and Climate, University of Minnesota, Saint Paul, Minnesota 55108, United States
[2]Department of Mechanical Engineering, University of Colorado Boulder, Boulder, Colorado 80309, United States

*Correspondence to*: Dylan B. Millet (dbm@umn.edu)

**Abstract.** We perform Observation System Simulation Experiments (OSSEs) with the GEOS-Chem adjoint model to test how well methane emissions over North America can be resolved using measurements from the TROPOspheric Monitoring Instrument (TROPOMI) and similar high-resolution satellite sensors. We focus analysis on the impacts of i) spatial errors in the prior emissions, and ii) model transport errors. Along with a standard scale-factor (SF) optimization we conduct a set of inversions using alternative formalisms that aim to overcome limitations in the SF-based approach that arise for missing sources. We show that 4D-Var analysis of the TROPOMI data can improve monthly emission estimates at 25 km even with a spatially biased prior or model transport errors (42–93% domain-wide bias reduction; R increases from 0.51 up to 0.73). However, when both errors are present, no single inversion framework can successfully improve both the overall bias and spatial distribution of fluxes relative to the prior on the 25 km model grid. In that case, the ensemble-mean optimized fluxes have a domain-wide bias of 77 Gg/d (comparable to that in the prior), with spurious source adjustments compensating for the transport errors. Increasing observational coverage through longer-timeframe inversions does not significantly change this picture. An inversion formalism that optimizes emission enhancements rather than scale factors exhibits the best performance for identifying missing sources, while an approach combining a uniform background emission with the prior inventory yields the best performance in terms of overall spatial fidelity—even in the presence of model transport errors. However, the standard SF optimization outperforms both of these for the magnitude of the domain-wide flux. For the common scenario in which prior errors are non-random, approximate posterior error reduction calculations for the inversions reflect the sensitivity to observations but have no spatial correlation with the actual emission improvements. This demonstrates that such information content analysis can be used for general observing system characterization but does not describe the spatial accuracy of the posterior emissions or of the actual emission improvements. Findings here highlight the need for careful evaluation of potential missing sources in prior emission datasets and for robust accounting of model transport errors in inverse analyses of the methane budget.



## 1 Introduction

Increases in atmospheric methane since the pre-industrial era have enhanced global radiative forcing by 0.97 W/m², making it the second-most important anthropogenic greenhouse gas after carbon dioxide (IPCC, 2013). However, the limited spatial coverage of observations has made it difficult to constrain emission distributions (Dlugokencky et al., 2011; Dlugokencky, 2020; Kirschke et al., 2013; Saunois et al., 2020). New space-based measurements from TROPOMI (TROPOspheric

Monitoring Instrument) provide near-global high-precision methane column observations at 7 km resolution, potentially filling this gap. In this study, we present Observing System Simulation Experiments (OSSEs) using a range of inversion strategies to explore the capabilities and limitations of high-resolution satellite-based column measurements for spatially resolving methane sources across North America.

Bottom-up methane emissions contain significant uncertainties. Recent global estimates for the 2008–2017 period range from

594–880 Tg/y (Saunois et al., 2020), with major disparities in spatial distribution. For example, while the total US anthropogenic methane flux in the Gridded Environmental Protection Agency inventory (GEPA: 29 Tg/y for 2012; (Maasakkers et al., 2016)) is within 15% of the corresponding estimate from the Emissions Database for Global Atmospheric Research v5 (EDGAR: 25 Tg/y for 2015; (Crippa et al., 2019; EDGAR v5, 2019)), these two datasets have a spatial correlation of just $R^2 = 0.01$ at 0.1°×0.1° resolution. Such spatial errors limit our ability to diagnose the reasons for model-measurement

disparities and thus hinder regional mitigation policies.

Atmospheric inversions are critical for testing and improving bottom-up methane flux estimates but carry their own uncertainties. Global top-down methane emission estimates for 2008–2017 range from 550–594 Tg/y and diverge substantially in their attribution of fluxes to source sectors, with differences up to a factor of 2 (e.g., 21–50 Tg/y for non-wetland natural emissions; (Dlugokencky et al., 2011; Saunois et al., 2020)). Such uncertainties also manifest on regional scales. For example,

recent top-down estimates for the US based on satellite (Greenhouse Gases Observing SATellite, GOSAT; Scanning Imaging Absorption Spectrometer for Atmospheric Chartography, SCIAMACHY) and in situ (tall tower, aircraft) measurements have varied between 30–45 Tg/y for different years, with differing source allocation (Maasakkers et al., 2021; Miller et al., 2013; Turner et al., 2015; Wecht et al., 2014a). Such disparities also manifest in other regions (Dlugokencky et al., 2011; Kirschke et al., 2013).

Prior spatial emission errors present one major barrier to top-down flux estimation. Inverse analyses commonly employ Bayesian scale factor (SF) optimization to improve flux estimates based on model-measurement concentration mismatches (Chen et al., 2018; Chen et al., 2021; Deng et al., 2014; Hooghiemstra et al., 2012; Jacob et al., 2016; Li et al., 2019; Maasakkers et al., 2021; Turner et al., 2015; Wecht et al., 2014a; Yu et al., 2021; Zhang et al., 2018). However, this approach fails where emissions are either missing entirely in the prior inventory or are too low to sufficiently adjust without incurring a prohibitive

cost-function penalty (Chen et al., 2018). In such cases the optimization will then tend to attribute the required emissions to



locations with higher prior emissions that require only modest adjustment and involve a smaller penalty (Jacob et al., 2016). Previous studies (e.g., Maasakkers et al., 2016; Turner et al., 2015) have further shown that prior spatial uncertainties in the sectoral allocation of emissions limit the accuracy of top-down source attributions. Employing normalized spatial surrogates (e.g., livestock distribution) as prior constraints can eliminate the dependence on bottom-up inventories (Michalak et al., 2004;

Miller et al., 2013), but the same limitations will apply given uncertainty in the spatial surrogates or variability in their relationship to fluxes.

Limited observational coverage has historically presented another major limitation to top-down methane analyses and exacerbates the prior dependencies outlined above. Ground-based networks provide a high-precision, temporally dense and long-term record of atmospheric methane concentrations at globally distributed sites (AGAGE, 2021; NOAA ESRL/GMD,

2021; WMO/WDCGG, 2021). However, these observations are spatially sparse compared to the heterogeneity of emissions. Airborne measurements (e.g., Davis et al., 2018; Gonzalez et al., 2021; Karion et al., 2015; Knox et al., 2019; Kort et al., 2008; Miller et al., 2013; Sheng et al., 2018b; Smith et al., 2017; Wofsy et al., 2018; Yu et al., 2020; Yu et al., 2021) expand this spatial footprint but only for discrete snapshots. Satellite measurements to date have generally also provided only limited coverage. For example, shortwave infrared (SWIR) methane measurements from GOSAT are separated in space by 260 km

(Kuze et al., 2016), while GHGSat observations are high-resolution (0.05° × 0.05°) but measure only a limited set of targets. Thermal infrared (TIR) measurements provide broad sampling but with limited sensitivity to methane emissions (Jacob et al., 2016).

High-resolution measurements from TROPOMI offer a major advance over earlier satellite observations for mapping methane emissions. Prior OSSEs have demonstrated this potential. For example, Wecht et al. (2014b) and Sheng et al. (2018a) found

that TROPOMI observations can provide comparable methane emission constraints as dedicated aircraft measurements spanning the same time intervals and regions. Other analyses have concluded that one week of TROPOMI methane observations is sufficient to resolve time-invariant fluxes at 30 km (Turner et al., 2018) and to achieve 100% error reduction over emission hotspots (Bousserez et al., 2016), while a single satellite overpass is able to monitor the 20 highest-emitting locations in the GEPA inventory (Jacob et al., 2016). However, the above work has focused primarily on resolving emission

magnitudes without explicitly considering the impacts of spatial errors.

Here, we apply the GEOS-Chem adjoint model in an OSSE framework to characterize the capabilities and limitations of TROPOMI and similar space-based sensors for resolving spatiotemporal patterns in methane emissions across local-to-regional scales. We perform an ensemble of synthetic inversions over North America, and specifically assess the ability of the observing system to spatially quantify heterogeneous emissions given limited confidence in their prior distributions. We further

evaluate multiple alternative inversion frameworks in terms of their strengths and weaknesses in this context relative to the standard and widely used SF approach.



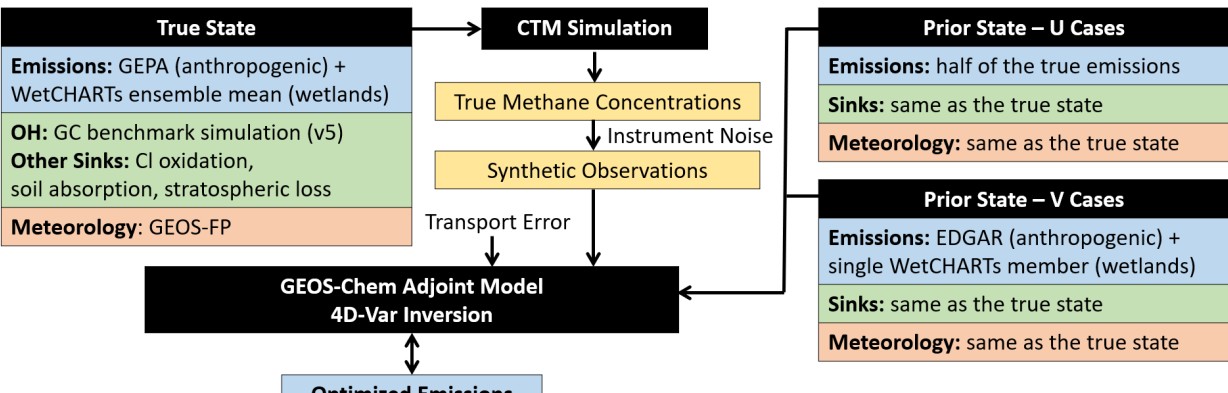

**Figure 1. Schematic illustrating the OSSE framework used here.**


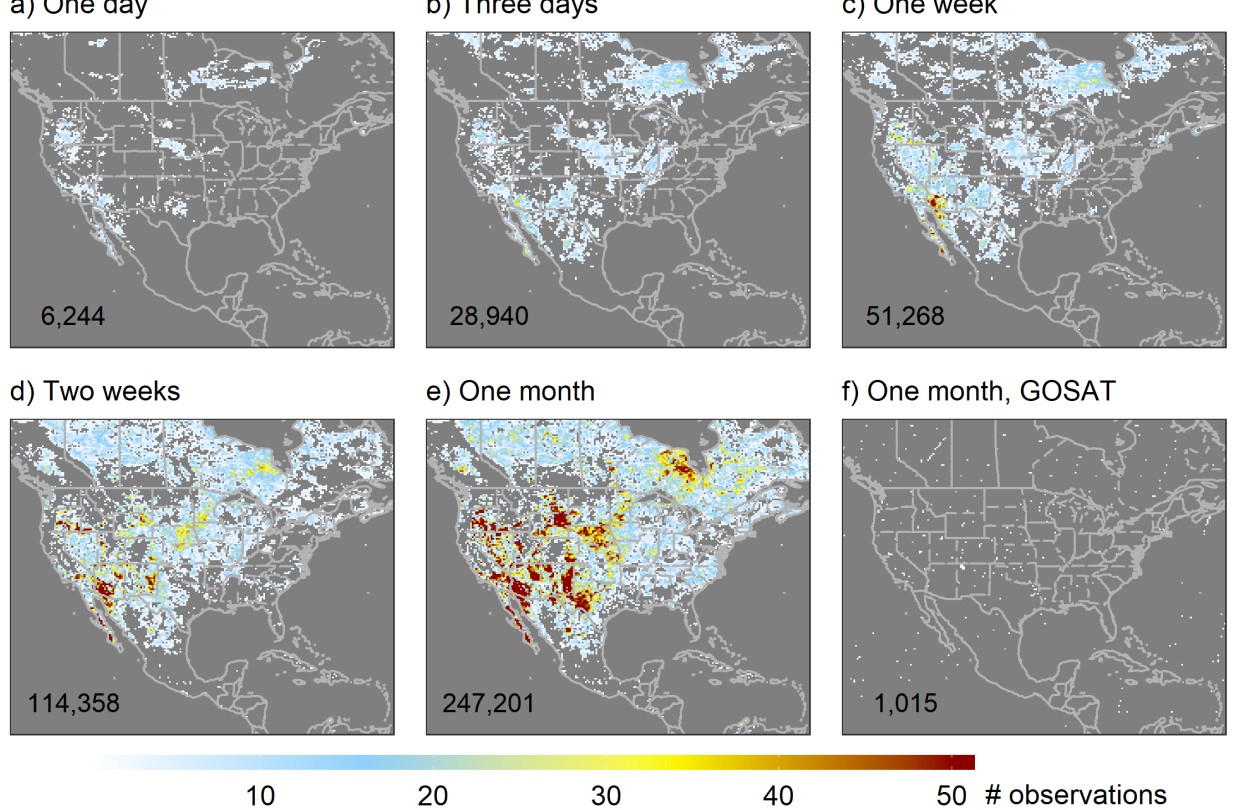

**Figure 2. TROPOMI sampling density at 0.25° × 0.3125° resolution (~25 km) for one-day to one-month observing intervals after filtering for data quality and clouds. GOSAT coverage (with no data filter) is shown for comparison. The total number of observations is labelled in each panel.**




**Table 1. Inversion frameworks**

| Emission bias | Inversion | | Framework[1] | Rationale |
|---|---|---|---|---|
| Spatially uniform emission errors | U-SF | Base-case SF[2] | $x = s \circ x_a$ | Explore reliability of optimized fluxes when spatial distribution is well-known in prior |
| Spatially varying emission errors | V-SF | Base-case SF | $x = s \circ x_a$ | Explore influence of spatial emission errors on base-case SF inversion |
| | V-flat | Flat prior | $x = x_{a\_ave} \, s$ | Identify constraints solely from TROPOMI without bottom-up knowledge |
| | V-AddBG | Background increment | $x = s \circ (0.5 \, x_a + 0.5 \, x_{a\_ave})$ | Identify missing sources |
| | V-OBSGuess | Observational guess | $x = s \circ (x_a + x_{ObsGuess})$ | Resolve and optimize emission hotspots |
| | V-EH | Enhancement | $x = x_{inc} \, s + x_a$ | Identify missing sources |
| Sensitivity inversions including transport error | *-*-T | | Same as U and V cases above | Assess the influence of model transport errors on methane source inversions |

[1]$x$: optimized emissions, $s$: scale factors, $x_a$: prior emissions, $x_{a\_ave}$: uniformly distributed prior emissions, $x_{ObsGuess}$: observationally informed initial guess, $x_{inc}$: emission increment.

[2]SF: scaling factor.

## 2 Methods

Figure 1 summarizes the OSSE framework employed here. We apply one month of synthetic TROPOMI observations over North America, with realistic instrument noise and sampling coverage, to evaluate the ability of different inversion frameworks
to recover the true distribution of methane sources. We test the impact of spatial biases by supplying each inversion with incorrect (but plausible) prior flux fields, both in the presence and absence of model transport errors. As is broadly the case for source inversions, the framework here is formally inconsistent with the best linear unbiased estimate (BLUE) 4D-Var problem. In general, we do not have an unbiased estimate of the prior so the solution will not be the BLUE. As a practical matter we thus seek to evaluate alternative inversion formalisms in terms of their ability to recover the true emissions in spite
of this limitation.





## 2.1 TROPOMI observations

TROPOMI is onboard Copernicus Sentinel-5 Precursor, a low-Earth polar-orbiting satellite launched in October 2017 with 13:30 local solar overpass time (LT). TROPOMI has a 2600 km swath width and provides near-global daily measurements at $7 \times 7$ km$^2$ nadir resolution in the shortwave infrared (SWIR) and $7 \times 3.5$ km$^2$ nadir resolution in the near infrared (NIR) (Hu et al., 2018). Methane columns are retrieved from NIR and SWIR spectral radiances with an estimated uncertainty of 1% due to instrument noise (0.6%) and forward model error (0.8%) (Hu et al., 2016; Hu et al., 2018; Lorente et al., 2021; TROPOMI Product User Manual, 2019).

We use synthetic observations for August 2018 in our analyses and apply standard data filtering procedures based on the actual TROPOMI retrieval quality parameters for clouds, spectral fit, albedo, aerosols, and viewing geometry (Table S1). Figure 2 shows the resulting data coverage for one-day to one-month intervals on the $0.25° \times 0.3125°$ analysis grid. Over 247,000 observations are available for August 2018 over North America, covering 66% of terrestrial grid cells, with the highest sampling density over the western US and northern Ontario. This level of data coverage is typical for TROPOMI: monthly overland coverage at $0.25° \times 0.3125°$ ranges from 42–79% between May 2018 and April 2019 when using the data selection criteria in Table S1 (Fig. S1). For comparison, Fig. 2f shows August 2018 data coverage for the GOSAT satellite sensor, with overland sampling density just 0.4% that of TROPOMI. Despite its significant sampling gaps, TROPOMI thus offers unprecedented new information for advancing scientific understanding of global methane sources and their spatiotemporal variability.

## 2.2 Chemical transport model and true state

Our OSSE analyses employ the GEOS-Chem (v11-2) chemical transport model (CTM) and its adjoint (v35) to optimize methane emissions on a $0.25° \times 0.3125°$ grid over North America (9.75°–60° N, 60°–130° W). Simulations use GEOS-FP meteorological fields and 5- and 10-minute timesteps for transport and emissions, respectively. Three-hourly dynamic boundary conditions are from simultaneous global simulations at $2° \times 2.5°$. Initial conditions are based on a global 25-year spinup for 2016 at $2° \times 2.5°$, followed by a two-week regional spinup over the nested domain at $0.25° \times 0.3125°$. As described next, inversions are performed for scenarios considering instrument error only and for scenarios considering both instrument error and model transport error. This permits comparison of these key observing system errors in terms of their impacts on solution accuracy.

Base case analyses include instrument error only, with the same transport scheme used to drive the adjoint model and to generate the true-state methane fields. Specifically, this relies on the transport implementation in v35 of the GEOS-Chem adjoint model, which includes fully instantaneous planetary boundary layer (PBL) mixing (Wu et al., 2007), a relaxed Arakawa-Schubert scheme (Moorthi and Suarez, 1992) for convection, and a multi-dimensional Flux-Form Semi-Lagrangian (FFSL) treatment of advection (Lin and Rood, 1996). The simulation also employs a six-cell ($0.25° \times 0.3125°$) buffer region





at the boundary between the global and nested simulation domains. We add 0.6% random error to the resulting methane column concentrations to represent instrument noise, and we apply the TROPOMI observation operator to the model output sampled instantaneously at the time and location of each satellite retrieval. In this way, the applied cloud coverage and other data filters

are consistent with the actual TROPOMI measurements.

Analyses that also incorporate model transport error rely on true-state tropospheric methane concentrations generated using v11-2 of the GEOS-Chem forward model. Transport here employs a non-local PBL mixing scheme (Lin and Mcelroy, 2010) and updated implementations of convection and advection (Zhang et al., 2021). The nested domain boundary uses a three-cell (rather than the six above) buffer region. Instrument error and the TROPOMI observation operator are then applied as before,

but to time-averaged (13:00–14:00 LT) rather than instantaneous tropospheric methane fields. The resulting TROPOMI methane columns have a mean root-mean-square error (RMSE) of 12 ppb relative to the base-case, with similar error contributions from transport and from the use of alternative emissions (Fig. S2). For comparison, Locatelli et al. (2013) reported a mean inter-model standard deviation of >15 ppb for surface concentrations between 10 CTMs (with identical emissions but differing transport) across a global ensemble of observing stations.

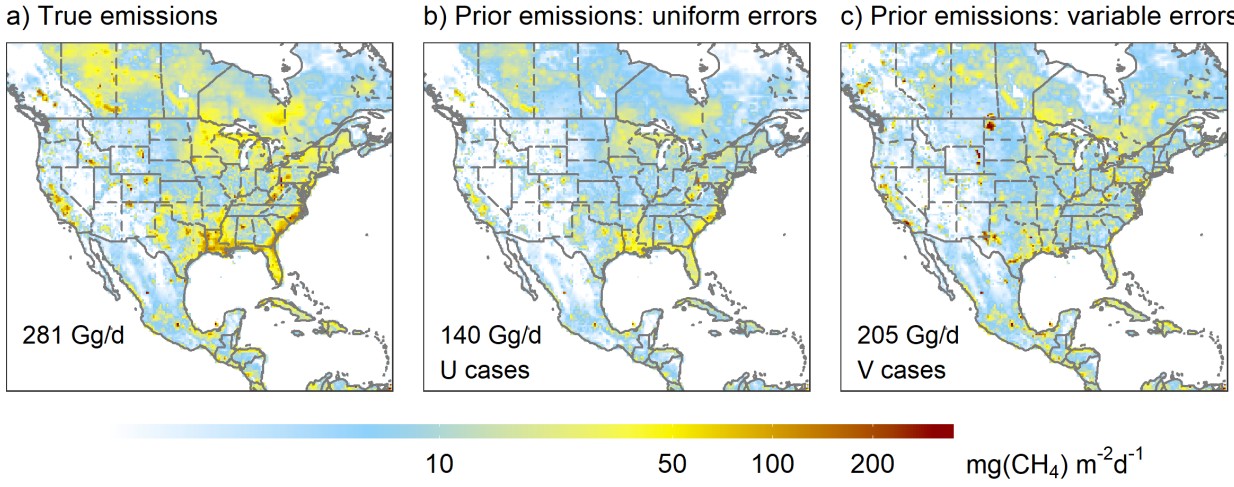


**Figure 3. True-state and prior emissions used in the OSSE analyses. The total emission for the North American domain is labelled in each panel.**

2.3 Methane sources and sinks

We use differing emission inventories to generate the true-state atmosphere and as prior for the inversions (Fig. 3). True-state anthropogenic fluxes are from the year-2012 Gridded EPA inventory (Maasakkers et al., 2016) over the US and from year-2012 EDGAR v4.3.2 (2017) elsewhere, totalling 124 Gg/d over the North America simulation domain. Wetland emissions use


the WetCHARTs ensemble mean (146 Gg/d for year-2017; (Bloom et al., 2017)), while biomass burning emissions are from the Quick Fire Emissions Dataset (QFED, 6 Gg/d for year-2017; (Darmenov and Silva, 2015)). Geological seep and termite

emissions follow Maasakkers et al. (2019) and Fung et al. (1991), respectively (together 5 Gg/d).

Prior emissions include scenarios with spatially uniform and spatially variable errors (designated as U and V cases, respectively; see Table 1). In the U cases, we scale the true-state emissions uniformly by 0.5×. This is a common OSSE approach and is informative when the prior emissions have strong spatial fidelity with the truth. However, when the spatial allocation of emissions is uncertain, as is frequently the case for methane, such analyses are likely to yield overly optimistic

results. We therefore also include prior scenarios based on an alternate set of inventories: EDGAR v4.3.2 for anthropogenic emissions (yielding a domain-wide anthropogenic source of 112 Gg/d), a single WetCHARTs ensemble member for wetlands (with $CH_4{:}C\ q_{10} = 1$, GLOBCOVER wetland extent, and a resulting flux of 80 Gg/d for year-2017), and QFED biomass burning emissions for a different year than in the true state (8 Gg/d for year-2018). The minor sources from geological seeps and termites are treated as before (5 Gg/d).

Figure 3 compares the true-state emissions with the above spatially perturbed prior. Across the domain, they differ by 76 Gg/d (27% of the true state), with large spatial disparities yielding an overall $R^2$ of 0.26. Major discrepancies are evident over oil and gas production regions (e.g., Bakken formation, Alberta oil sands), wetlands (e.g., Hudson Bay Lowlands, US south coast), and agricultural areas (e.g., California Central Valley, Upper Midwest).

Methane sinks in the model include oxidation by the hydroxyl radical (OH, 90% of the total simulated removal over the nested

domain), computed using archived monthly fields from a full-chemistry simulation (v5-07-08). Stratospheric oxidation contributes a further 6% and is computed based on NASA Global Modeling Initiative monthly loss frequencies (Murray et al., 2013). Other minor sinks include soil absorption (3%; following Fung et al. (1991)) and oxidation by chlorine (2%; following Sherwen et al. (2016)).

2.4 Optimization framework

We apply the GEOS-Chem adjoint model (Henze et al., 2007) to optimize the total methane flux ($\boldsymbol{x}$) in each 0.25° × 0.3125° grid cell via iterative reduction of the following cost function:

$$J(\boldsymbol{x}) = (\boldsymbol{x} - \boldsymbol{x_a})^T \mathbf{S_a^{-1}}(\boldsymbol{x} - \boldsymbol{x_a}) + \gamma\big(\boldsymbol{y} - F(\boldsymbol{x})\big)^T \mathbf{S_{obs}^{-1}}(\boldsymbol{y} - F(\boldsymbol{x})) \tag{1}$$

where $F(\boldsymbol{x})$ are the model-predicted methane columns, $\boldsymbol{y}$ are the synthetic TROPOMI observations, $\boldsymbol{x_a}$ are the prior emissions, and $\mathbf{S_a}$ and $\mathbf{S_{obs}}$ are respectively the prior and observational error covariance matrices. We employ a regulation parameter ($\gamma$)

to avoid overfitting, defined based on one-week sensitivity inversions with uniformly biased prior emissions (Fig. S3). We


then scale $\gamma$ for other time windows according to the number of observations ($\gamma = 8.1, 1.6, 1.0, 0.5,$ and $0.2$, for 1 day, 3 days, 1 week, 2 weeks, and 1 month, respectively).

By default, minimization of $J(x)$ in the GEOS-Chem adjoint proceeds through derivation of grid-level SFs $s$ that are then applied as $x = s \circ x_a$. Eq. 1 can thus be equivalently expressed in terms of $s$ rather than $x$, in which case $\mathbf{S_a}$ describes relative
rather than absolute errors. We first explore inversion performance in this framework, including the impacts of spatial emission errors and model transport errors. We subsequently evaluate four alternative inversions as candidates to address shortcomings of this SF approach; these are summarized in Table 1 and detailed in Sect. 4.

Prior error covariances for the GEPA anthropogenic emissions include magnitude and displacement components and are computed following Maasakkers et al. (2016). Those for wetlands are estimated as the standard deviation of the WetCHARTs
model ensemble (Bloom et al., 2017). Other emissions assume an error standard deviation of 50%, consistent with previous work (Maasakkers et al., 2019; Sheng et al., 2018b; Turner et al., 2015; Wecht et al., 2014a; Zhang et al., 2018). The above components, weighted by the corresponding flux amount, are added in quadrature to obtain the diagonal elements of $\mathbf{S_a}$. The resulting prior errors have a median value of 67% for the uniform-bias inversions (which employ GEPA as anthropogenic prior) and 142% for the others (which employ EDGAR v4.3.2). Finally, we employ an exponentially decaying 200 km
correlation length-scale to populate the off-diagonal elements of $\mathbf{S_a}$, in-line with previous studies (200–500 km (Monteil et al., 2013; Wecht et al., 2014a; Yu et al., 2021)).

Observational error covariances are prescribed as the relative residual standard deviation of the column mismatch between the true-state synthetic observations and the prior simulations over a 2°×2° moving window (Heald et al., 2004). We impose on the derived values a lower limit of 60 ppb$^2$, corresponding to the 0.25 quantile of the overall error distribution. The resulting
observing system errors average 9 ppb (range: 8–29 ppb) and mainly reflect instrument noise. This is similar to observational error estimates for previous methane inversions using data from TROPOMI (e.g., 11 ppb; (Zhang et al., 2020)) and GOSAT (e.g., 13 ppb; (Maasakkers et al., 2019)) and is therefore an appropriate representation for our OSSE analyses.

We derive posterior errors and degrees of freedom for signal (DOFS) for each of the inversions using a gradient-based randomization approach (Bousserez et al., 2015). The posterior error matrix $\mathbf{S_{opt}}$ is given by:

$$\mathbf{S_{opt}} = \left(\mathbf{S_a^{-1}} + \mathbf{H^T S_{obs}^{-1} H}\right)^{-1} = \left(\mathbf{S_a^{-1}} + \overline{\nabla J(x_a)\,\nabla J(x_a)^T}\right)^{-1} \tag{2}$$

where $\mathbf{H}$ is the forward model operator and $\nabla J(x_a)$ is the cost function gradient at $x = x_a$. The $\overline{\nabla J(x_a)\,\nabla J(x_a)^T}$ term is computed from an ensemble of cost function gradients, each relying on synthetic data that have been perturbed with random noise based on the error characteristics of the original dataset. The DOFS are then computed as the trace of $\mathbf{I} - \mathbf{S_{opt} S_a^{-1}}$. Our computed DOFS converge at approximately 100 ensemble members.







**Figure 4. Differences between the initial-guess emissions and the true state, and between the optimized emissions and the true state, for each inversion framework. Labels inset indicate the change in domain-wide bias and spatial correlation with respect to the true fluxes achieved through each optimization. More details are provided in Table 1 (inversion specifications) and Fig. S4 (initial-guess and optimized emissions). Inversions shown here do not include model transport errors; Fig. S6 shows results when such errors are included.**




**Figure 5. Inversion performance by source size and sector in terms of root-mean-square error (RMSE) and spatial correlation coefficient (R) relative to the true fluxes. Results from base-case inversions (instrument error only) are shown for a) all grid cells, b) small sources (< 50 mg/m²/day), c) large sources (≥ 50 mg/m²/day), d) small missing sources (prior < 10 mg/m²/day; truth ∈ [10, 50] mg/m²/day), e) large missing sources (prior < 10 mg/m²/day; truth ≥ 50 mg/m²/day), f) fossil fuel, g) livestock, h) other anthropogenic, i) wetland, and j) other natural emissions. Panels k–t show results including model transport error. The total true fluxes are indicated in each panel.**





**Figure 6. Inversion performance in terms of domain-wide emission bias for (top to bottom): all grid cells, small sources (< 50 mg/m²/day), large sources (≥ 50 mg/m²/day), small missing sources (prior < 10 mg/m²/day; truth ∈ [10, 50] mg/m²/day), large missing sources (prior < 10 mg/m²/day; truth ≥ 50 mg/m²/day), and for fossil fuel, livestock, other anthropogenic, wetland, and other natural emissions. Panel a shows results for synthetic observations subject to instrument error only, while panel b shows results that also include model transport errors.**

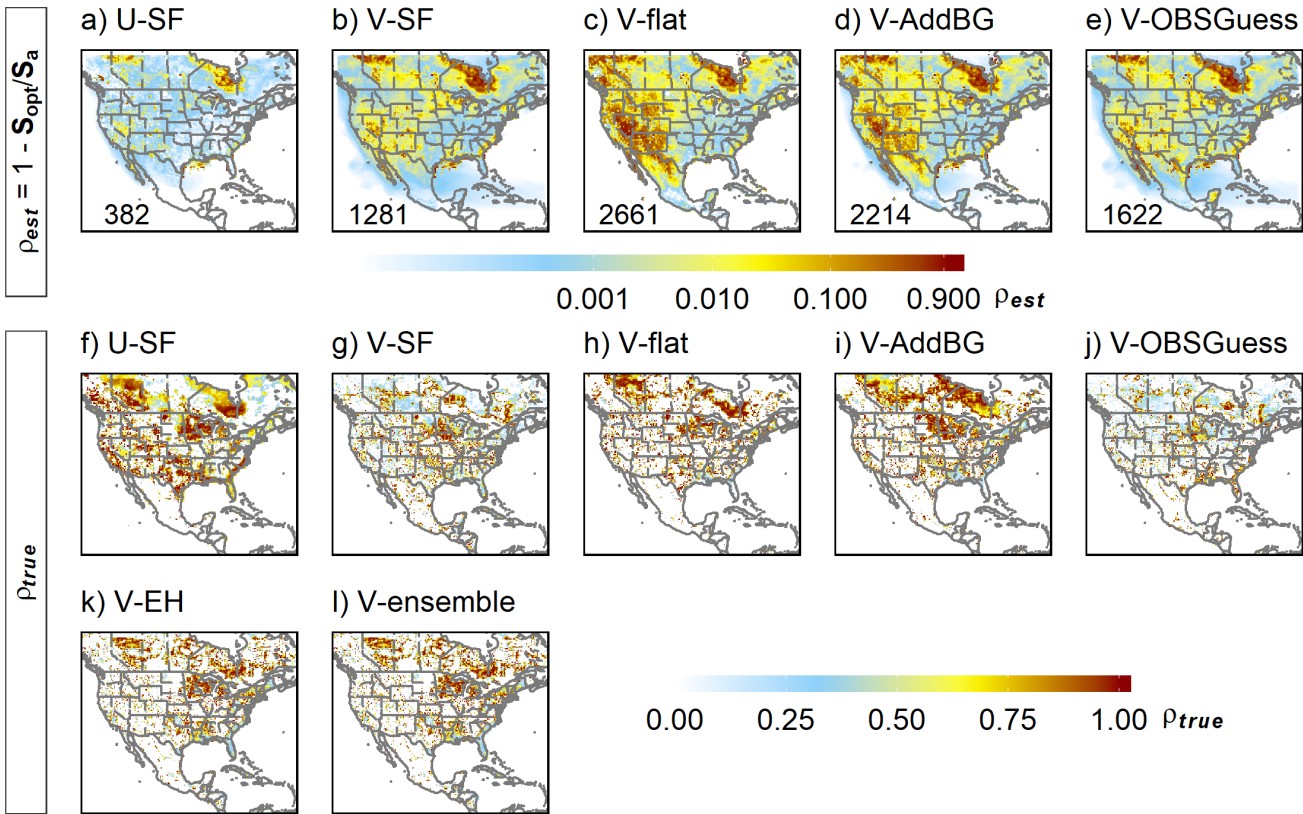

**Figure 7. Posterior error reduction (top row; degrees of freedom for signal are labelled inset) and actual grid-level emission improvement (bottom rows) for each inversion framework. Table 1 provides the inversion details for each case.**

We compute these information content metrics because they are commonly used for evaluating satellite instrument capabilities (Bousserez et al., 2016; Sheng et al., 2018a). However, posterior error reduction estimates can only match the emission improvements if the prior emissions are unbiased, which is not usually the case. For this reason, we compare the posterior error reduction $\boldsymbol{\rho}_{est}(i) = 1 - \mathbf{S}_{opt}(i,i)/\mathbf{S}_a(i,i)$ estimated as above against the actual grid-level emission improvement:

$$\boldsymbol{\rho}_{true}(i) = 1 - \left(\hat{\boldsymbol{x}}(i) - \boldsymbol{x}_{true}(i)\right)^2 / \left(\boldsymbol{x}_a(i) - \boldsymbol{x}_{true}(i)\right)^2 \tag{3}$$

where $\hat{\boldsymbol{x}}(i)$ and $\boldsymbol{x}_{true}(i)$ are respectively the optimized and true emissions for grid cell $i$. For computational reasons we employ only the diagonal elements of $\mathbf{S}_a$ in the calculation of $\boldsymbol{\rho}_{est}$; an evaluation using a random subset of grid cells suggests that this approximation alters the results by <25%.





## 3 Role of prior errors in biasing inversions

### 3.1 Inversions with spatially uniform prior biases

Our results show that in the absence of model transport errors, a one-month standard SF inversion of TROPOMI observations can effectively minimize a uniform prior emission bias while maintaining spatial fidelity with the truth (U-SF inversion; Table

1). Figure 4 (a–b) shows that for this scenario the prior bias of 140 Gg/d is reduced by 61% (to 54 Gg/d), while spatial correlation with the true fluxes decreases only slightly from R = 1 to R = 0.93. The U-SF inversion successfully improves the prior estimates for both small (< 50 mg/m$^2$/d) and large (≥ 50 g/m$^2$/d) sources, in all cases maintaining R > 0.8 with the true fluxes and decreasing model RMSE at the 25 km grid resolution (Fig. 5). Domain-wide flux estimates for these source categories are improved by 46% and 95%, respectively (Fig. 6). Partitioning of emissions between anthropogenic and natural

sources (by scaling the derived total fluxes to the prior grid-level source fractions) is also accurate, with R > 0.9 and decreased RMSE for every case except livestock (which had low error to begin with) and improved domain-wide flux accuracy (Figs. 5 and 6).

Despite this strong overall performance, we do see an influence from the prior emission distributions on the inversion results. The spatial correlation between the derived SFs and the prior emissions is R = 0.58, whereas the true solution (SF = 2 for all

grid cells) would have no such correlation. This reflects a tendency for SF inversions to over-correct large sources while under-correcting small sources. If the actual prior emission errors were random and normally distributed (i.e., no mean bias), the impacts of this tendency would manifest equally everywhere and would not lead to spatially coherent SF-prior correlations. Here, we employ a uniform prior emission bias, which (like most real scenarios) breaks the assumption of unbiased Gaussian emission errors. SF inversions are widely used despite this limitation, and we see here that the approach broadly succeeds

under a uniform-bias scenario even with the tendency for large-source overcorrection.

The U-SF inversion has DOFS = 382, with derived posterior error reductions that reflect the TROPOMI spatial sampling density for this month (Fig. 7). However, this computed error reduction $\rho_{est}$ has no meaningful spatial correlation with the actual emission improvement $\rho_{true}$ (R = 0.07). This reflects the fact that the posterior error reductions and DOFS contain no information on where the prior emissions are actually in error and can therefore be improved. For a scenario where the prior

emissions had random and normally distributed disparities relative to the truth, areas with the largest computed posterior error reduction would also tend to have the greatest emission improvement—since those locations would have the strongest observational constraints. DOFS and error reduction analyses are thus useful for general observing system characterization, but do not describe the spatial accuracy of posterior emissions or the actual emission improvements for realistic scenarios where the real prior errors are non-random.

The imposition of model transport errors does not strongly degrade the above performance. In this case (U-SF-T inversion; Table 1), the domain-wide optimized emission magnitude for North America is no less accurate than before (in fact slightly



more so; Fig. 8b). The spatial distribution of the derived emissions, while less precise than in the case with perfect model transport, maintains high spatial fidelity with the truth (R > 0.8; Fig. 8b). As we will see later, the same is not true when spatially varying prior errors are present.


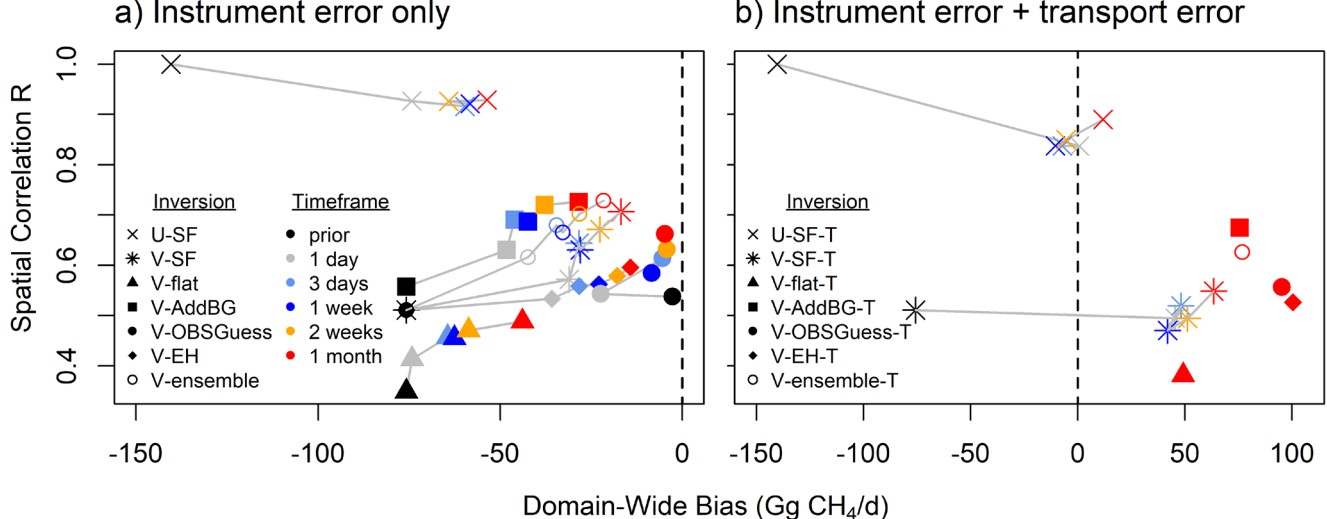

**Figure 8. OSSE performance for timescales ranging from one day to one month (colors) as a function of inversion framework (symbols). Table 1 provides the inversion details for each case. Panel a shows results for synthetic observations subject to instrument error only, while panel b shows results that also include model transport errors (panel b shows only the one-month results for the alternative formalisms).**


### 3.2 Inversions with spatially varying prior biases

When spatially varying biases are present in the prior emissions but transport errors are absent, the standard SF approach (V-SF inversion; Table 1) can still successfully minimize the domain-wide flux bias (78% reduction, from 76 to 17 Gg/d) and has

moderate success in recovering the true spatial distribution of emissions (R increases from 0.51 to 0.71). However, unlike the strong performance seen across all sources in the uniform bias case, Figs. 5 and 6 show that the V-SF inversion reduces the domain-wide emission biases for both small and large sources, but fails to improve the spatial allocation of large sources (RMSE and R improve by just 0.2% and 0.02, respectively). Comparing Fig. 4 (c–d) and 5, we see that the V-SF inversion successfully corrects erroneous hotspots that are overestimated in the prior (e.g., Bakken and Permian shales) but lacks the

ability to identify missing sources (e.g., wetland emissions in Alberta, the Hudson Bay Lowlands, and the US south coast). Thus, given the presence of spatially varying prior biases, the SF tendency to over-correct large sources and under-correct small sources discussed above now manifests in systematic ways that bias the derived fluxes for particular regions and sectors.





The V-SF inversion has DOFS = 1281, higher than the U-SF case due to the larger prior error estimates for EDGAR versus GEPA. However, as with the U-SF inversion, the estimated posterior error reductions $\rho_{est}$ and the actual emission improvements $\rho_{true}$ have no significant spatial correlation at the 95% confidence level—for the reasons discussed earlier. As an example, the computed posterior error reductions shown in Fig. 7b show large uncertainty decreases over the Hudson Bay Lowlands (with high observational density), but the derived fluxes over this region do not in fact improve towards the truth.

Combining model transport errors with the spatially varying prior emission errors substantially worsens the SF inversion performance. We then obtain an over-correction of the domain-wide flux and a resulting positive bias (23%) that is almost as large as the prior negative bias, with the optimization also failing to meaningfully improve the spatial accuracy of the prior emissions (Fig. 8b). As a result, the inversion has little ability to resolve sectoral sources: with the exception of livestock, none of the posterior sectoral fluxes improve over their prior estimates in terms of RMSE or correlation, and only wetland sources are improved in terms of domain-wide bias (Figs. 5 and 6). As discussed in Sect. 5, extending the duration of the analysis period does not significantly improve the situation (Figs. 8b). This finding aligns with a previous ensemble analysis of surface observations (Locatelli et al., 2013), in which the optimized fluxes varied by 23–48% regionally and up to 150% at the grid level (2.5° × 3.75°) depending on model transport scheme. These disparities clearly point to model transport error as one of the primary reasons behind the large spread in top-down methane source attributions (Locatelli et al., 2013).

In summary, while TROPOMI is ground-breaking in providing high-precision, high-resolution methane observations daily and on a global basis, the combination of i) spatial errors in prior emission estimates with ii) model transport errors continues to limit inversion performance. Careful evaluation of potential missing sources in the prior inventories, along with rigorous evaluation of model transport errors, is critical for robust inverse analyses.

## 4 Alternative approaches to mitigate impacts of spatially varying prior errors

The impacts of prior spatial errors discussed above present a general challenge to inverse analyses. The fact that SF-based emission adjustments of a given magnitude incur less penalty over high-emission grid cells can be problematic for sources with poorly known spatial distributions, such as wetlands. For methane, with its long atmospheric lifetime, such issues are compounded by model transport biases, since the inversion cost function is then heavily weighted by downwind observations with their accumulating errors.

We examine here four alternative inversion strategies in terms of their capacity for addressing these limitations. As summarized in Table 1 and described below, these include three SF inversions in which the initial guesses are modified from the standard prior (V-flat, V-AddBG, and V-OBSGuess) and a fourth inversion that optimizes absolute emission enhancements (V-EH) rather than SFs. As before, we evaluate inversion performance in each case based on one month of synthetic TROPOMI observations both in the absence and presence of model transport errors.





### 4.1 Flat-Prior inversion (V-flat): Good spatial performance for small sources but poor constraints on large sources

Our first alternative inversion (V-flat; Table 1) employs initial-guess emissions that are distributed uniformly among model

land grid cells, with the aim of resolving spatial flux patterns entirely from the TROPOMI observations while also addressing the inability of the SF inversion to recover missing sources. The initial domain-wide flux magnitude is consistent with that in the standard prior, and the inversion penalty term is computed with respect to the revised initial guess.

Figure 4 (e–f) shows that in the absence of transport error the V-flat inversion reduces the total prior emission bias by 42% (from -76 Gg/d to -44 Gg/d, the weakest performance among all inversions) and recovers a significant portion of the true

spatial variance (R = 0.49; however, this is still lower than for the original prior emissions). The optimization captures some broad patterns in the true fluxes, with higher emissions in the eastern US and over Canada (Fig. S4), but does not resolve key finer-scale features. In particular, the optimized fluxes yield no improvement for large sources, either in terms of spatial distribution or emission magnitude (Figs. 5 and 6). The inversion performs well at recovering the spatial distribution of small sources (RMSE decreases by 54%; R increases from 0.39 to 0.71), and it captures their combined source magnitude to within

345   15%.

When transport error is included (V-flat-T, Table 1), the magnitude of the domain-wide emission bias increases from -44 to +49 Gg/d. The V-flat-T inversion is still able to improve the RMSE (by 25%) for small sources but dramatically overestimates their domain-wide magnitude (by 63%) to compensate for a 78% underestimate of large sources (Fig. 6). Given the V-flat inability to constrain large sources and poor overall spatial correlation with the truth, we move on to examine other inversion

approaches below.

### 4.2 Background-Increment inversion (V-AddBG): Best spatial performance

Our second alternative framework (V-AddBG; Table 1) occupies a middle ground between the V-SF inversion and the V-flat inversion. Here we construct an initial guess field as the mean of the standard prior and the flat initial guess used above, with the aim of incorporating prior knowledge while also enabling the SF inversion to recover missing sources.

Figure 4 (g–h) shows that in the absence of model transport error, the V-AddBG inversion successfully reduces the regional mean bias from the prior value of -76 Gg/d to -28 Gg/d. While this overall bias correction is not as strong as the standard V-SF approach (63% vs. 78%), the V-AddBG inversion yields higher spatial correlation with the truth across the domain (R = 0.73 vs. 0.71). Sectoral performance combines the strengths of the V-SF and V-flat inversions. Like V-flat, V-AddBG has strong spatial performance for small sources (reducing the RMSE by 56% and increasing R from 0.39 to 0.75; Fig. 5), while

mitigating the V-flat tendency to overestimate small sources and underestimate large sources. Like V-SF, V-AddBG can effectively correct false hotspots (e.g., Bakken and Permian shales) while being better able to resolve missing sources (e.g.,



Hudson Bay Lowlands). Some missing sources (e.g., US Gulf Coast) are still not resolved due to limited observational coverage.

Transport errors (V-AddBG-T inversion; Table 1) increase the domain-wide bias in the derived fluxes from -28 Gg/d to +76
Gg/d. However, this framework still provides the best spatial accuracy among all approaches examined, with R = 0.67 relative to the true fluxes even with transport error. It has better spatial performance than V-SF-T for every source category except wetlands, where the two are comparable (Fig. 5). For several emission categories where the standard V-SF-T solution is spatially inferior to the prior (small sources, large sources, fossil fuel, livestock), V-AddBG-T delivers meaningful improvements (Fig. 5). Overall, we find that V-AddBG provides the best spatial fidelity across all inverse approaches, but the
domain-wide bias improvements are in general not as strong as the standard V-SF approach.

4.3 Observational-Guess inversion (V-OBSGuess): Exploiting long-term observations to identify missing sources

Our third alternative framework (V-OBSGuess; Table 1) exploits the TROPOMI observations themselves to derive initial-guess emissions that enable SF-based recovery of spatially heterogeneous and missing sources. Specifically, we use one year of synthetic TROPOMI data (generated as above for the true-state atmosphere and averaged to the model resolution (Sun et
al., 2018)) to identify locations $i$ with high methane column $\Omega_i$ but low prior emissions $x_i$ based on the following index (see Fig. S5):

$$\beta_i = (x_{max} - x_i)/x_{max} \times \Omega_i/\Omega_{max} \tag{4}$$

For locations exceeding the 0.8 quantile for this index we linearly scale $\beta_i$ to a corresponding prior emission increment. The scaling relationship is derived from scaled-emission sensitivity simulations with the resulting increments capped at 50 mg/m$^2$/d. The role of atmospheric transport means that this conversion is merely an approximation, but it is suitable for our
purposes as input for a source inversion—and in particular, addresses the failure of standard SF inversions for scenarios with a near-zero prior where a source actually exists. The initial guess flux field derived in this way reduces the regional emission bias in the standard prior by 97% (from -76 Gg/d to -3 Gg/d) and slightly improves its spatial fidelity to the truth (from R = 0.51 to 0.54), thus providing an improved starting point for the 4D-Var optimization.

Figure 4 (i–j) shows that in the absence of transport error, the V-OBSGuess inversion maintains the low domain-wide bias
present in its initial guess; however, this does reflect some compensation between large-source underestimates and small-source overestimates (Fig. 6). The spatial accuracy of emissions is improved from R = 0.54 to R = 0.66. Specific improvements are apparent for locations with erroneous hotspots in the prior (e.g., Bakken and Permian shales) as well as those with missing sources (e.g., Hudson Bay Lowlands). Emissions are likewise improved over southern US coastal regions, where observational coverage is low for this month, thanks to the revised initial guess that employs a full year of data. V-OBSGuess yields moderate
spatial performance for sector-specific emissions (Fig. 5), outperforming V-SF slightly for missing sources but otherwise not





exhibiting a particular benefit for any specific emission category. Overall, when omitting transport error, the V-OBSGuess approach provides the lowest domain-wide bias of any inversion, with comparable spatial performance to the standard V-SF approach.

However, this approach is highly sensitive to model transport error (V-OBSGuess-T; Table 1). In its presence, the domain-
wide emission bias increases from -5 Gg/d to 95 Gg/d, which is worse than the original prior. The spatial allocation of the derived emissions is not substantially improved over the revised initial guess (R = 0.56 versus 0.54) or the original prior (R = 0.51). Therefore, while V-OBSGuess can achieve a low regional-mean bias and strong spatial fidelity given accurate model transport, it is strongly influenced by meteorological errors.

### 4.4 Emission enhancement inversion (V-EH): Recovering large missing sources

Finally, we optimize emission enhancements rather than SFs (V-EH; Table 1). In this case, the prior error covariances in Eq. 1 are computed with respect to absolute flux increments rather than with respect to the prior emissions; the inversion thus has increased flexibility to add emissions everywhere regardless of the local prior. We employ an increment of $x_{inc}$ = 10 kg/grid/min and optimize emissions as $x = x_{inc} \, s + \, x_a$. For context, approximately half of the domain-wide prior emissions are contributed by grid cells with emissions greater than $x_{inc}$. We further set a lower bound for the scaled increment (of $-x_a$)
to avoid negative fluxes, and prescribe zero fluxes for ocean grid cells outside of offshore oil/gas production fields.

Figure 4 (k–l) shows that in the absence of model transport error, the V-EH inversion reduces the regional emission bias from -76 Gg/d to -14 Gg/d (by 81%) while improving the spatial correlation from R = 0.51 to 0.60. It succeeds at removing some of the false hotspots present in the prior (i.e., Bakken and Permian shales), but incorrectly smears those corrections spatially due to atmospheric transport. The V-EH inversion exhibits the best overall performance for missing sources (prior emissions
< 10 mg/m$^2$/d; Figs. 5 and 6): the RMSE and domain-wide bias are reduced by 25% and 62%, respectively, for small missing sources (with R increasing from 0 to 0.39 with respect to the truth), and by 16% and 22%, respectively, for large missing sources (but with no clear correlation improvement in this case).

Transport errors significantly degrade the optimized fluxes in this framework (V-EH-T; Table 1), increasing the domain-wide bias from -14 Gg/d to 101 Gg/d and reducing the spatial fidelity to R = 0.53. The latter is only a slight improvement over the
prior (R = 0.51) and lower than is obtained with the standard V-SF-T approach (R = 0.55). V-EH-T still yields the best overall improvement for large missing sources, but in other cases the performance is mixed (Figs. 5 and 6).

### 4.5 Summary and ensemble inversion performance

The inversion approaches explored above offer distinct advantages and disadvantages, which we summarize below.





1) In the absence of model errors (Fig. 5), V-flat and V-AddBG exhibit strong spatial performance for small sources (<50 mg/m²/d, representing 97% of grids cells and 70% of total emissions), but overestimate their emissions while underestimating those for large sources. For large sources (≥50 mg/m²/d, 3% of grid cells, 30% of total emissions), all inversions except V-flat yield modest improvements. V-EH performs best at recovering missing sources (e.g., for wetlands) while the V-AddBG results generally have the highest spatial correlation with the truth. At the sectoral level, the V-SF, V-AddBG and V-OBSGuess inversions are all able to improve over the prior estimates for fossil fuel emissions (characterized by false hotspots in the prior) and livestock emissions (which have a spatially accurate prior). The alternative approaches allow more spatial flexibility in source allocation than the standard SF inversion, but the trade-off is a greater propensity to introduce spurious sources.

2) When subject to model errors, the V-SF-T, V-AddBG-T and V-OBSGuess-T inversions all overcorrect large sources, while the V-flat-T and V-EH-T inversions create some spurious sources to compensate for transport biases. The V-AddBG-T inversion is the only framework able to reduce the grid-level emission RMSE despite transport errors, and it achieves the highest spatial R across all inversions (Fig. 5). However, it is unable to reduce the domain-wide emission bias given the transport errors imposed here. Conversely, the V-SF-T and V-flat-T inversions are able to improve the overall emission bias in the presence of transport errors, but they fail to improve their spatial accuracy. Short-lived species that are less influenced by transport error would likely yield better performance.

3) In all cases, the posterior error reductions $\rho_{est}$ derived via Eq. 2 have no significant correlation with the actual emission improvements $\rho_{true}$ (Fig. 7).

Together, the above inversions provide a range of possible emission solutions to fit the TROPOMI data. For optimizations using real data, analyzing the mean and spread across inversion frameworks provides a way to combine those various constraints and to diagnose robust aspects of the solution (e.g., Tarantola, 2006; Yu et al., 2021). Here, in the absence of transport error, the ensemble mean solution reduces the domain-wide bias from -76 Gg/d to -22 Gg/d (by 72%; Fig. 4) with higher spatial accuracy (R increases from 0.51 to 0.73, RMSE is reduced by 33%) than any individual inversion. In the presence of transport errors, the ensemble mean offers performance intermediate among the individual cases for both domain-wide emission bias (77 Gg/d vs. -76 Gg/d in the original prior) and overall spatial fidelity (R increases from 0.51 to 0.63, RMSE is reduced by 11%).

Restricting the analysis to grid cells where the derived emission adjustments have consistent sign across all inversions yields a larger improvement over the prior. In the instrument-error-only cases, the spatial fidelity across these grid cells (totalling 127 Gg/d, 45% of total emissions and 31% of the grid cells) improves from R = 0.21 to R = 0.53 with a 46% reduction in the grid-level RMSE, and total emission magnitude bias improves by 79% (from -52 Gg/d to -11 Gg/d). In the presence of both instrument and model transport errors, the spatial fidelity of these consistent grid cells (totalling 147 Gg/d, 52% of total



emissions and 40% of the grid cells) improves from R = 0.25 to R = 0.50 with a 27% grid-level RMSE reduction, but total
emission magnitude bias only improves by 2% (from -61 Gg/d to 60 Gg/d).

## 5 Influence of inversion timeframe on solution accuracy

The above analyses are all based on one month of synthetic observations and performed on the 0.25° × 0.3125° model grid.
Below we extend the analysis to alternate timeframes to further evaluate the impact of data coverage on inversion performance.

We find that in the absence of both spatially varying prior emission errors and model transport errors (U cases; Table 1), a
single day of TROPOMI measurements can reduce a domain-wide emission bias by 47% (from -140 Gg/d to -74 Gg/d, Fig.
8a) while maintaining high spatial fidelity with the truth (R > 0.9). Extending the inversion timeframe to 3 days reduces the
bias by an additional 10% (to -60 Gg/d), but longer timeframes beyond this offer no additional benefit (our true emissions in
the OSSE do not vary on a sub-monthly timescale). When model transport error is present but prior emissions are still spatially
accurate, we again find similar performance across one-day to one-month timeframes, with strong bias reduction and spatial
accuracy in each case (Fig. 8b, U-SF-T, Table 1). Therefore, given spatially reliable bottom-up inventories, TROPOMI can
constrain methane emissions at high time resolution and resolve day-to-day temporal variability even in the presence of realistic
model transport errors.

Spatial errors in the prior change this picture. In that case, when model transport error is absent (V cases; Table 1), both the
emission bias and spatial fidelity continue to improve as the inversion timeframe increases from one day to one month (Fig.
8a). Given accurate transport, additional TROPOMI observations thus allow all the inversions to progressively correct prior
spatial inaccuracies. However, the combination of transport errors with spatial emission errors in the prior strongly
compromises inversion performance in all cases. Except for V-AddBG, we see at best marginal spatial improvements with
increasing observational coverage (Fig. 8b). Furthermore, the domain-wide bias progressively worsens as the inversion
timeframe increases, and after one month the bias in the derived fluxes has comparable magnitude to that in the prior. While
longer inversion windows benefit from increased data coverage, this comes at the cost of accumulating transport error that can
degrade performance. In such cases, multiple short inversions, rather than a single long inversion, may be preferable.

## 6 Conclusions and implications

In this paper we examine three factors that limit the accuracy of top-down methane source estimates: i) observational coverage,
ii) spatial inaccuracies in prior emission estimates, and iii) model transport accuracy. The TROPOMI satellite sensor provides
unprecedented, high-density and high-precision measurements of atmospheric methane columns over land, representing a
major step forward in addressing the first of these constraints. We employ here a series of OSSE experiments to evaluate a



range of inversion approaches in terms of their ability to spatially resolve methane emissions from high-coverage satellite sensors such as TROPOMI given the remaining limiting factors above.

The widely used SF-based inversion approach can be problematic for sources with poorly known spatial distributions, since adjusting grid cells with missing or near-zero sources in the prior may incur a prohibitive cost-function penalty. The required emissions are then unfortunately allocated to higher-emission locations. We examine four alternate inversion strategies that aim to alleviate this issue. Three use a revised initial guess to allow SF-based recovery of missing sources (V-flat: flat prior; V-AddBG: adds a background emission to the standard prior; V-OBSGuess: uses satellite data to pre-allocate missing sources)

while the fourth optimizes emission enhancements (V-EH) rather than scale factors. The V-EH inversion performs best at resolving missing sources, whereas V-AddBG has the best performance in terms of the overall spatial fidelity of the solution— even in the presence of model transport errors. However, the standard V-SF approach yields better domain-wide bias reduction when model transport errors are present. The spread and mean across the ensemble solutions help in identifying robust aspects and uncertainties in the optimized flux patterns. For example, grid cells in which the emission corrections have consistent sign

across the ensemble members exhibit improved grid-level RMSE reduction (here, 27% in the presence of transport error) compared to other grid cells (-8%, degraded performance).

We find that 4D-Var source optimization based on TROPOMI observations can provide robust constraints on monthly methane emissions at 25-km resolution, even when provided spatially incorrect prior emissions or in the presence of significant model transport error. However, performance is substantially degraded when both of these errors are present. Then, only one of the

inversion frameworks is able to improve upon the prior spatial distribution of emissions (V-AddBG-T; R increases from 0.51 to 0.67 and RMSE reduces by 21%), but it is unable to reduce the domain-wide emission bias. The two inversion frameworks that successfully reduce the prior bias in this scenario (V-SF-T and V-flat-T; 16–36% bias reduction) are unable to improve the spatial distribution (grid-level RMSE). In many cases spurious emission adjustments are derived to compensate for the transport errors. Increasing observational coverage through longer-timeframe inversions does not resolve the situation,

providing only a modest spatial improvement but with progressively worsening domain-wide bias due to accumulating transport errors.

We show through the OSSE analysis that the computed error reduction has no meaningful correlation with the actual emission improvements that are obtained in the inversions. This arises because, in general, the true prior emission disparities are not randomly distributed with zero mean (as is formally required in the best linear unbiased estimate, or BLUE, 4D-Var problem)

but rather have coherent spatial patterns associated with specific source types and regions. While often applied for observing system characterization, this approximate information content analyses should not be used to assess inversion accuracy or the spatial reliability of derived fluxes.

Findings here show that improving the spatial accuracy of bottom-up methane emission estimates is one key need for advancing top-down source assessments—for example through wetland extent surveys, better assessment of the environmental drivers of
fluxes, and precise facility-level information for livestock, fossil fuel, and industrial facilities. However, even with best efforts in these areas, the temporally sporadic nature of certain fluxes, combined with uncertainties in sectoral partitioning and in emission drivers, will inevitably lead to some bottom-up spatial biases. Such challenges provide the main motivation for pursuing top-down approaches in the first place.

Fundamental advancement will therefore require both the minimization of model transport errors and explicit representation
of such errors in inverse analyses. On-going efforts to improve CTM representation of inter-hemispheric transport, convection, and boundary layer mixing offer promise for reducing the influence of model transport errors in future inverse analyses (Lin and Mcelroy, 2010; Patra et al., 2011; Saito et al., 2013). Including model error terms in the cost function for optimization, for example via weak-constraint inverse modelling, can alleviate the perfect-model dependence of standard (strong-constraint) inverse approaches and would improve inversion results for long-lived tracers such as methane (Stanevich et al., 2020, 2021).
As such developments progress, current inverse analyses of space-borne methane measurements require careful evaluation of possible missing sources and of model transport errors, along with thoughtful assessment of the potential for multiple viable solutions.

**Code and data availability**

TROPOMI data is publicly available at http://www.tropomi.eu/data-products/level-2-products. The GEOS-Chem adjoint code
is available at http://adjoint.colorado.edu:8080/gcadj_std.git. The modified code used here is archived at https://doi.org/10.13020/g5xc-nj81.

**Author contributions**

XY performed the 4D-Var inversions. DBM and DKH supervised this study. All authors contributed to the interpretation of results. XY and DBM wrote the manuscript. All authors reviewed and edited the manuscript.

**Competing interests**

The authors declare that they have no conflict of interest.



**Acknowledgements**

We thank Hansen Cao, Colin Harkins, and Nicolas Bousserez for helpful discussions, and thank Kelley C. Wells and Timothy J. Griffis for their contributions to the GEM (Greenhouse Emissions in the Midwest) project. The GEM project is supported by NASA's Interdisciplinary Research in Earth Science program (IDS Grant #NNX17AK18G) and by the Minnesota Supercomputing Institute. XY acknowledges support from a NASA Earth and Space Science Fellowship (Grant #80NSSC18K1393).

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
