# Peer review of "How well can inverse analyses of high-resolution satellite data resolve heterogeneous methane fluxes? Observation System Simulation Experiments with the GEOS-Chem adjoint model (v35)"

_Geoscientific Model Development, 2021_

## Author Comment (AC1)

**Review #1 Evaluations:**

**We thank the reviewer for the thoughtful comments. Reviewer comments are provided below in black with our responses in blue.**

Yu et al present a series of Observation System Simulation Experiments (OSSEs) to test how well methane emissions from North America can be resolved using TROPOMI satellite data. They consider different inversion approaches in the presence of errors in the prior spatial distribution, and model transport error. Overall, the manuscript is well-written and presented although I have a few comments and questions that might be addressed.

**Thank you for the insightful review. We have revised the draft based on the reviewer's specific suggestions as below.**

**General comments:**

The authors analyse biases between posterior and truth at two scales, grid-scale and whole domain. In general, whole-domain performance exceeds grid-scale performance. This is not surprising given the sensitivity of observations to any single grid cell vs the domain as a whole and errors in the spatial distribution. Analyses are presented for individual source sectors but I am curious as to how the spatial overlap between sectors affects this. Could the authors comment on spatial scales at which posterior biases of the source total approach that of the domain-wide bias? i.e. through successive coarsening of the posterior grid scale solutions. Some discussion of certain regions is provided (e.g. Bakken, Permian), but it might be useful to provide a more direct analysis over these key emission areas.

**Thank you for the suggestion. Figure C1 below shows the spatial correlation of the inversion results with the true fluxes at successively coarser posterior grid scale solutions.**

[Figure]

**Figure C1. Spatial correlation of inversion results against the true fluxes with successive coarsening of the grid scale.**

**We indeed see that the solutions improve in terms of their spatial correlation to the truth. However, the improvements largely arise from a corresponding improvement in the prior fluxes (⊕), rather than being achieved through the inversions.**

To address this point in the manuscript, we have added the following plot of the results at 4°×5° to the supporting information (Fig. S6) for comparison with the native grid solution in Figure 8.

[Figure]

Figure 8. OSSE performance for timescales ranging from one day to one month (colors) as a function of inversion framework (symbols). Table 1 provides the inversion details for each case. Panel a shows results for synthetic observations subject to instrument error only, while panel b shows results that also include model transport errors (panel b shows only the one-month results for the alternative formalisms).

[Figure]

Figure S6. Same as Figure 8, but with prior and posterior results degraded to 4°×5° horizontal resolution.

We have also added the following discussion to the main text:

"Given the combined effects of imperfect model transport, prior spatial errors, and limited observational coverage, one might expect better inversion accuracy when evaluated at coarser resolution rather than on the native 0.25° × 0.3125° model grid. When aggregating the solutions to 4° × 5° we indeed obtain higher correlation with the true fluxes for all cases with spatially variable prior errors (Fig. S6). However, this result mainly arises because the prior itself is more spatially accurate at

this scale: the degree of improvement actually achieved through the inversion is no better than at fine resolution. Overall, neither the use of alternative timeframes nor of spatial aggregation changes our finding that inversion experiments can improve both the spatial distribution and magnitude of fluxes if either spatially variable prior errors or transport errors are present—but not when both errors are present."

Regarding the reviewer's question about key emission regions, we considered additional new text for inclusion on this point. However, we find that it does not add substantive new information to the manuscript beyond what was already presented in the context of source sectors and spatial coverage. So, we have chosen not to add this to the paper.

One key feature of North America is the concentration of CH4 sources (and prior errors) in the East, but the majority of TROPOMI retrievals are in the West. Is there any value in analysing the skill of each inversion in those areas which are well-observed by the data separately to those areas that are not?

Thank you for the suggestion. Figure C2 shows the emission improvement from prior to posterior (in terms of root-mean-square error [RMSE]) as a function of observational coverage. We do not see a consistent relationship between data coverage and emission improvement. For example, the mean improvements are comparable between grid cells with n = 1 and those with n = 25. While there is less spread in the improvements (i.e., fewer instances with significant degradation) for grid cells with the highest coverage (e.g., SD = 4.3 for n ≥ 18), we do not believe this is necessarily a robust effect of sample size since we also see the highest spread (e.g., SD =42.9 for n = 13-17) at intermediate coverage rather than at low coverage (SD = 9.7 for n ≤ 10).

[Figure]

Figure C2. Absolute posterior error minus absolute prior error, in terms of RMSE, as a function of number of observations per grid. Results are shown for inversions with a) instrument error only and b) instrument error and transport error.

In the main text, we evaluated the impact of data coverage on inversion performance in two ways. First, we computed the posterior error reduction (equation 2 in the main text) using the gradient-based randomization approach. This posterior error reduction is determined by the data coverage, the observational error estimates, and the prior error estimates. We now directly express this connection to data coverage in section 2.4. We then discussed the comparisons between the posterior error estimates and the actual grid-level emission improvements in sections 3 and 4. Second, we evaluated the impact of data coverage by considering alternative inversion timeframes from 1 day to 1 month in section 5. When model transport error is absent, a longer timeframe helps to correct spatial emission errors; however, this is not generally the case in the presence of model transport error.

The discussion on posterior error reduction and correlation with the true error seems specific to the approximation of the posterior error used in the 4D var approach. This should perhaps be made clearer in the abstract and other points in the text.

As suggested, we have now clarified that these uncertainties are derived from the gradient-based randomization approach: in the abstract, in the Figure 7 caption, and in sections 3 and 6.

**Specific comments**

P8, Section 2.4: Does the approach also optimize boundary conditions? For an OSSE this obviously isn't really the focus, but the transport error simulations change the treatment of boundary conditions which presumably would affect the results if boundary conditions are not optimized.

We did not optimize boundary conditions, and this is now stated in section 2.4.

P9, L204: What is the impact of the imposed 200 km prior error correlation length-scale on derived emissions at grid-scale? To what extent would this affect your results, assuming the true error correlation is different?

To address this question, we performed new sensitivity inversions varying the spatial correlation length scale. Results are now discussed in the text as follows:

"Finally, we employ an exponentially decaying 200 km correlation length scale to populate the off-diagonal elements of $S_a$, in-line with previous studies (200–500 km (Monteil et al., 2013; Wecht et al., 2014a; Yu et al., 2021)). Sensitivity tests varying this length scale yield broadly similar results: derived scale factors for one-week V-SF inversions using 100 km, 200 km, and 400 km correlation lengths are spatially correlated to R = 0.82-0.94 and have <1% domain-wide emission differences."

P9, L207: Do the authors account for off-diagonal observation error covariances? The lack of description suggests not. That may be an issue for the high-resolution TROPOMI data. Furthermore, do the model-transport error simulations exhibit spatial error correlation structures? Could the inversion performance be improved by accounting for these in the off-diagonals of the observation error covariance matrix?

We did not account for off-diagonal observation error covariances, as this capability is not currently available in the GEOS-Chem adjoint model. The model transport error indeed exhibits spatial error correlation structure, as shown in the existing Figure S2 (reproduced as follows).

We agree with the reviewer that accounting for off-diagonal covariance estimates may improve future satellite-based inversions and have added discussion on this point to the manuscript:

"The current version of the GEOS-Chem adjoint model assumes that the observational error covariance matrix is diagonal. However, model transport errors have clear spatial correlation structure (Fig. S2), and future work accounting for off-diagonal observational errors may thus improve performance."

[Figure]

**Figure S2.** Methane column concentration differences for 2018-08-01 resulting from the individual model transport errors employed in the OSSE (see Sect. 2 for details). Shown are differences incurred from: a) using 6 versus 3 buffer grid cells at the domain boundary; b) averaging over 13:00-14:00 LT versus sampling the model instantaneously at the satellite overpass time; c) employing non-local versus full PBL mixing schemes; d) alternate convection and tropopause treatments; and e) all model transport errors. Shown for comparison are f) the column differences that arise from employing the true versus spatially biased prior emissions. The root-mean-square errors (RMSE) relative to the true fluxes are labeled in each panel.

P9. L211: Even quality controlled TROPOMI XCH4 data seem to be impacted by systematic errors due to albedo and other considerations (Lorente et al, 2021). So perhaps the uncertainties here are an optimistic representation?

We thank the reviewer for pointing this out and have revised the draft as suggested.

"The 9 ppb estimate is in-line with and slightly smaller than observational error estimates for previous methane inversions using data from TROPOMI (e.g., 11 ppb; (Zhang et al., 2020)) and GOSAT (e.g., 13 ppb; (Maasakkers et al., 2019)); it is therefore an appropriate representation for our OSSE analyses. Note that any systematic measurement errors (Lorente et al., 2021) are inherently not accounted for in our framework and would need separate correction."

P13, Fig 7: Could you define what $\rho_{est}$ and $\rho_{true}$ are in the figure caption?

**We have revised the label and caption for Figure 7 as suggested.**

P14, L 265: "This reflects a tendency…" Doesn't this also reflect that the satellite data are not sensitive to small sources?

**We have revised this text as suggested:**

**"This reflects a tendency for SF inversions to over-correct large sources while under-correcting small sources (along with the fact that the satellite data themselves are less sensitive to small sources)."**

P18, Section 4.3: Is the prior actually independent of the data in the V-obsguess case? In effect aren't the data being used twice to optimize the emissions? Once to create the prior and secondly in the assimilation.

**This is correct and we have now clarified this in the text as follows.**

**"This inversion thus optimizes emissions in two steps, using the synthetic observations to first spatially identify missing sources then to optimize grid-level emission magnitudes."**

P19, L398: …"it is strongly influenced by meteorological errors" Is it any more influenced by errors than other methods, and if so why?

**Yes, it is more influenced by meteorological errors than other methods. We now discuss this point in the main text as suggested:**

**"Therefore, while V-OBSGuess can achieve a low regional-mean bias and strong spatial fidelity given accurate model transport, its flexibility in spatially correcting emissions also makes it more sensitive to meteorological errors than other approaches."**

P20, L421: "all inversions except V_flat yield modest improvements". Improvements relative to what?

**We have clarified this as suggested.**

**"For large sources (≥50 mg/m2/d, 3% of grid cells, 30% of total emissions), all inversions except V-flat yield modest improvements (e.g., 0.2-12% reduction in grid-level RMSE relative to the true fluxes)."**

P22, L503: "This arises because…the true prior emissions disparities are not randomly distributed with zero mean" If I understand correctly, this is a direct result of the way the error matrix is calculated in Eq. 2. Does this mean that this caveat or mismatch is only applicable to the 4D-var approach and the posterior error approximation? Would one expect a different outcome using an exact Bayesian rather than approximate approach to error estimation?

**We now address this point in the main text as follows.**

**"We show through the OSSE analysis that the computed error reduction (approximated via gradient-based randomization) has no meaningful correlation with the actual emission improvements that are obtained in the inversions. This arises because, in general, the true prior emission disparities are not randomly distributed with zero mean (as is formally required in the best linear unbiased estimate, or BLUE, 4D-Var problem) but rather have coherent spatial patterns associated with specific source types and regions. The same issue would also apply had the posterior errors instead been derived exactly via an analytical Bayesian solution."**

P23, L508-513: Presumably if bottom-up estimates were improved sufficiently we wouldn't need top-down approaches. Just a comment, but correcting errors in bottom-up distributions seems like a basic requirement of top-down approaches.

We very much agree with this point and include it in the second to last paragraph of the main text.

"Findings here show that improving the spatial accuracy of bottom-up methane emission estimates is one key need for advancing top-down source assessments—for example through wetland extent surveys, better assessment of the environmental drivers of fluxes, and precise facility-level information for livestock, fossil fuel, and industrial facilities."

**Review #2 Evaluations:**

We thank the reviewer for the constructive review. Reviewer comments are provided below in black with our responses in blue.

The authors conduct Observing System Simulation Experiments (OSSEs) to assess the ability of TROPOMI observations of atmospheric methane columns for constraining monthly methane emissions at the 25 km scale through inverse analyses. I think the paper is within the scope of GMD and is well-written. I recommend publishing the paper after considering my comments below:

Thank you for the positive comments. We have revised the draft based on the reviewer's specific suggestions as follows.

**General comments:**

The analyses of inversion performance are done over the whole North America. As the authors conducted high-resolution (~25km) inversions, I think it would be more interesting to focus on regional performance. For example, southeast US includes large sources from oil/gas, agriculture, and wetlands, while the number of observations is very limited there as shown in Fig. 2. What would inversion performance look like in this region?

Thank you for the suggestion. We considered additional new text for inclusion on the regional performance. However, we find that it does not add substantive new information to the manuscript beyond what was already presented in the context of source sectors and spatial coverage. So, we have chosen not to add this to the paper.

[Figure]

Figure C2. Absolute posterior error minus prior error, in terms of RMSE, as a function of number of observations per grid. Results are shown for inversions with a) instrument error only and b) instrument error and transport error.

We further evaluated the impact of data coverage on inversion performance as follows.

Figure C2 shows the emission improvement from prior to posterior (in terms of root-mean-square error [RMSE]) as a function of observational coverage. We do not see a consistent relationship between data coverage and emission improvement. For example, the mean improvements are comparable between grid cells with n = 1 and those with n = 25. While there is less spread in the improvements (i.e. fewer instances with significant degradation) for grid cells with the highest coverage (e.g., SD = 4.3 for n ≥ 18), we do not believe this is necessarily a robust effect of sample size since we also see the highest spread (e.g., SD =42.9 for n = 13-17) at intermediate coverage rather than at low coverage (SD = 9.7 for n ≤ 10).

In the main text, we evaluated the impact of data coverage on inversion performance in two ways. First, we computed the posterior error reduction (equation 2 in the main text) using the gradient-based randomization approach. This posterior error reduction is determined by the data coverage, the observational error estimates, and the prior error estimates. We now directly express this connection to data coverage in section 2.4. We then discussed the comparisons between the posterior error estimates and the actual grid-level emission improvements in sections 3 and 4. Second, we evaluated the impact of data coverage by considering alternative inversion timeframes from 1 day to 1 month in section 5. When model transport error is absent, longer timeframe help to correct spatial emission errors; however, this is not generally the case in the presence of model transport error.

**Specific comments:**

L43-44: "a spatial correlation of just R2". Do you mean coefficient of correlation (R) or coefficient of determination (R2)? In Section 3 the authors start to use R (e.g. Fig 4). I would suggest the authors use either R or R2 throughout.

Thank you for catching this. We now use R in all cases, as suggested.

L118-119: How do you generate your synthetic observations? Do you do a forward model simulation and then sample XCH4 at the locations of TROPOMI observations?

This is correct and is explained in the text as follows:

"We add 0.6% random error to the resulting methane column concentrations to represent instrument noise, and we apply the TROPOMI observation operator (averaging kernel and prior methane profiles) to the model output sampled instantaneously at the time and location of each satellite retrieval."

L134-135, L142-143: It looks like instrument error includes only random error. What about systematic instrument error?

As is generally the case in 4D-Var optimization, our inversion framework includes only random errors. Any systematic errors need to be identified and corrected separately and prior to the inversion. We have clarified this in-text as follows.

"The 9 ppb estimate is in-line with and slightly smaller than observational error estimates for previous methane inversions using data from TROPOMI (e.g., 11 ppb; (Zhang et al., 2020)) and GOSAT (e.g., 13 ppb; (Maasakkers et al., 2019)); it is therefore an appropriate representation for our OSSE analyses.

**Note that any systematic measurement errors (Lorente et al, 2021) are inherently not accounted for in our framework and would need separate correction."**

L143: Perhaps add a bit more details about TROPOMI observation operator?

**This has now been clarified as described in our response to the reviewer's comment on L118-119.**

L168: "This is a common...". Citations?

**We have added citations as suggested.**

**"This is informative when the prior emissions have strong spatial fidelity with the truth and is a common OSSE approach (e.g., Bousserez et al., 2016; Sheng et al., 2018; Turner et al., 2018)."**

L186: Equation (1). Tikhonov regularization for an inverse problem commonly includes the regularization parameter (γ) in regularization term (i.e. prior ). Is there a particular reason to put γ in the observation-model term in the cost function? In the supplementary, γ is determined by L-curve. I assume you are referring to Hansen (2005) L-curve criterion which is based on Tikhonov regularization.

**We apply γ to the observation-model terms in the cost function to be consistent with previous studies such as Qu et al. (2021, 10.5194/acp-21-14159-2021) and Lu et al. (2021, 10.5194/acp-2021-671). We indeed used the L-curve from Hansen (1993) and we have added a citation accordingly.**

In addition, what do you mean "Cost function - Obs" and "Cost function - Prior" in Fig. S3. Please write down the equations you use. Also, please set x/y-axis as log scale for Fig. S3a

**We have revised Figure S3 as suggested, and defined the relevant equations in the caption.**

[Figure]

**Figure S3. Cost function analysis and determination of the regularization parameter $\gamma$ based on one-week inversions with spatially uniform prior errors. Panel a) shows the L curve comparing the prior and observational deviation terms in the cost function as a function of $\gamma$, following the method in Hansen and O'Leary (1993). As shown in Eq. 1 of the main text, the prior term is given by $(x - x_a)^{\mathrm{T}} S_a^{-1} (x - x_a)$ and the observational term is given by $(y - F(x))^{\mathrm{T}} S_{obs}^{-1} (y - F(x))$. Panel b) shows the prior term divided by the total cost function computed at $\gamma = 1000$ ($J_{1000}$, where the solution is mostly determined by the observations; blue line), the observational term divided by the total cost function computed at $\gamma = 0$ ($J_0$, where the solution is solely determined by the prior; red line), and the sum of the blue and red lines (in grey).**

L214: Equation (2). Where is the regularization parameter γ?

**Thank you for pointing out this typo. We have corrected equation 2 as follows.**

$$\mathbf{S_{opt}} = (\mathbf{S_a^{-1}} + \gamma\mathbf{H^T S_{obs}^{-1} H})^{-1} = \left(\mathbf{S_a^{-1}} + \gamma\overline{\nabla J(x_a)\,\nabla J(x_a)^T}\right)^{-1}$$

L368-369: "meaningful improvements"? Perhaps be more quantitative and specific about "improvements" throughout the whole text.

**We have revised the corresponding text as follows.**

**"For several emission categories where the standard V-SF-T solution is spatially inferior to the prior (small sources, large sources, fossil fuel, livestock), V-AddBG-T delivers meaningful improvements— improving the grid-level RMSE by up to 27% and the spatial correlation to the truth by up to 76% (Fig. 5)."**

**We also modified the following paragraph to address this point:**

**"For large sources (≥50 mg/m2/d, 3% of grid cells, 30% of total emissions), all inversions except V-flat yield modest improvements (e.g., 0.2-12% reduction in grid-level RMSE relative to the true fluxes)"**

L471: what do you mean "accumulating transport error"?

**We have now clarified this point in the text, as follows.**

**"While longer inversion windows benefit from increased data coverage, this comes at the cost of accumulating transport error as atmospheric enhancements are related to emissions farther and farther upwind."**